# Concurrent Analysis of Antioxidant and Pro-Oxidant Activities in Compounds from Plant Cell Cultures

**DOI:** 10.3390/biotech14040091

**Published:** 2025-11-14

**Authors:** Marcela Blažková, Ľubica Uváčková, Mária Maliarová, Jozef Sokol, Jana Viskupičová, Tibor Maliar

**Affiliations:** 1Department of Biology and Biotechnology, Faculty of Natural Sciences, University of Ss. Cyril and Methodius in Trnava, Nám. J. Herdu 2, 917 01 Trnava, Slovakia or macrela.blazkova@nppc.sk (M.B.); lubica.uvackova@ucm.sk (Ľ.U.); 2National Agricultural and Food Centre, Hlohovecká 2, 951 41 Lužianky, Slovakia; 3Department of Chemistry and Environmental Sciences, Faculty of Natural Sciences, University of Ss. Cyril and Methodius in Trnava, Nám. J. Herdu 2, 917 01 Trnava, Slovakia; maria.maliarova@ucm.sk (M.M.); jozef.sokol@ucm.sk (J.S.); 4Centre of Experimental Medicine SAS, Institute of Experimental Pharmacology and Toxicology, Slovak Academy of Sciences, Dúbravská Cesta 9, 841 04 Bratislava, Slovakia; jana.viskupicova@savba.sk

**Keywords:** antioxidant activity, pro-oxidant activity, secondary metabolites, plant cell culture

## Abstract

Oxidative stress reflects an imbalance between pro-oxidants and antioxidants arising from physiological or environmental factors. Here, we applied our previously developed in situ microplate method for the simultaneous determination of antioxidant and pro-oxidant activities to compounds produced by plant cell cultures in vitro. The primary aim was to evaluate the added value of these compounds, which are widely used as additives in food, cosmetic, and pharmaceutical products. The secondary aim was to assess whether a predominance of pro-oxidant activity could limit their biotechnological production. Thirty-three compounds known to be produced by in vitro cultures (polyphenolic acids, flavonoids, quinones, alkaloids, etc.) were tested, and the pro-oxidant–antioxidant balance index (PABI) was calculated. Sixteen compounds showed measurable activities with DPPH_50_/FRAP_50_ values below 2 mM. Within this set, rosmarinic acid exhibited pronounced pro-oxidant behavior, whereas gallic acid, chlorogenic acid, and the anthocyanin cyanidin showed higher antioxidant potency and favorable PABI values. Such compounds may deliver added benefits when incorporated into food or cosmetic products and are unlikely to limit production in cell culture.

## 1. Introduction

Oxidative stress refers to an imbalance between the generation of reactive oxygen/nitrogen species and the capacity of a biological system to neutralize these reactive forms. Substances capable of mitigating or reducing oxidative stress through any mechanism are classified as antioxidants [1]. Antioxidants are widely used in the pharmaceutical, cosmetic, and food industries, where, among other mechanisms, they help stabilize lipid matrices [2,3]. It has been reported that up to 80% of the global population relies on plant-based products not only as food sources but also for healthcare purposes [4]. Plant-based products are rich in phytochemicals, which are utilized as nutraceuticals [5], cosmeceuticals [6] and phytopharmaceuticals [7].

The global phytochemical market was estimated at USD 7 billion in 2024, with projections indicating growth to USD 17 billion by 2034 [8]. This substantial value reflects the beneficial effects attributed to phytochemicals, traditional medicine, public trust, and a widespread belief in their efficacy with minimal side effects, depending on the mode of administration.

However, not all natural sources of valuable phytochemicals are domesticated or amenable to agronomic production systems [9]. Several factors drive the need for alternative, biotechnological production methods, including low natural yield due to slow growth rates, limited availability of plant resources, and low or highly variable concentrations of the target compound(s), often influenced by a range of biotic and abiotic factors.

Plant cell or organ cultures offer an independent, environmentally sustainable, and controllable source of bioactive compounds—provided that production levels are industrially relevant [9]. Establishing such cultures for industrial-scale production is complex and typically involves (i) the selection of high-yielding plant cell lines or organ cultures; (ii) optimization of cultivation conditions (e.g., elicitation, immobilization, permeabilization, biotransformation); (iii) scale-up processes (bioreactor design, operation optimization, and cultivation strategy); and (iv) the industrial production phase.

The literature documents many examples of phytochemical production using plant in vitro cultures, which can be grouped by chemical class and biosynthetic origin [10]. Representative classes of plant-derived secondary metabolites include alkaloids, terpenoids, steroids, quinones, and phenylpropanoids. Alkaloids are compounds containing at least one nitrogen atom in their molecular structure. This group may include potential production of acridines, furoquinolines, galanthamine, harringtonines, isoquinolines, lobeline, piperidine, quinolizidines, indole alkaloids, isoquinoline alkaloids, thebaine, and trigonelline or tropane alkaloids. Terpenoids, produced via the mevalonate biosynthetic pathway, include compounds such as artemisinin, cucurbitacins, ginsenosides, meroterpenes, paclitaxel and other taxanes, sesterterpenes, thapsigargin, ursane triterpenoids and withanolides. Steroids are characterized by a fully reduced or modified cycloperhydrophenanthrene ring system. This category comprises brassinolide, bufadienolides, catasterone, digoxin, digitoxins, ecdysteroids, physodine, ouabain, helleborin, scillaridine, and various steroidal glycosides and lactones. Quinones are defined by a diketone configuration within a cyclic skeleton with various geometries. These include benzoquinones such as thymoquinone; naphthaquinones such as β-lapachone, plumbagin, shikonin, juglone; and anthraquinones such as emodin, aloe emodin, chrysophanol, rhein or phenanthrenequinones. Finally, phenylpropanoids, named after their biosynthetic pathway, comprise anthocyanins, coumarins, polyphenolic acids, eugenol, various flavonoids, lignans, proanthocyanidins, stilbenes, tannins, and related compounds [10]. These plant secondary metabolites play a significant role in the therapy of various serious diseases, including oncological disorders [11,12,13], bacterial and infectious diseases [14,15,16,17], inflammatory conditions such as rheumatoid arthritis [18,19,20], diabetes [21,22,23], and neurological disorders such as depression and Parkinson’s disease [24,25].

The primary objective of this pilot study is to evaluate potential differences in the balance between antioxidant and pro-oxidant activities among compounds produced through in vitro plant biotechnological systems. Elucidating these differences may provide valuable insights into their prospective applicability, while also indicating that compounds exhibiting a predominance of pro-oxidant activity could constitute intrinsic self-limiting factors within the biosynthetic processes of their own production systems.

## 2. Materials and Methods

### 2.1. Chemicals

Chemicals and solvents, including 2,2-diphenyl-1-picrylhydrazyl radical (DPPH^●^), 2,2-diphenyl-1-picrylhydrazine (DPPH), 2,4,6-tris(2-pyridyl)-s-triazine (TPTZ), FeCl_2_·4H_2_O, FeCl_3_·6H_2_O, sodium acetate, 96% (*v*/*v*) ethanol, and acetic acid, and analytical standards—tannic acid, rhamnetin, α-tocopherol, gallic acid, 1,2,3,4,6-penta-o-galloyl-β-d-glucose/5GGL/, chlorogenic acid, cyanidin hydrochloride, kaempferol, rosmarinic acid, sinapic acid, arbutin, 1,4-dithiothreitol, syringic acid, polydatin, luteolin, hesperetin, 3-hydroxyflavone, biochanin A, scopolamine, podophyllotoxin, berberine, paclitaxel, rhein, olivetol, plumbagin, phloridzin (phlorizin), *p*-coumaric acid, aesculin, hesperidin, icariin, lapachol, shikonin, and purpurin were purchased from Merck/Sigma (Burlington, MA, United States).

### 2.2. Microplate Method

Antioxidant and pro-oxidant properties were assessed using microplate-based DPPH and FRAP assays. Both assays employed identical concentrations (0.4 mM) of the key reagents DPPH, TPTZ, and FeCl_3_. A 0.4 mM DPPH^●^ solution was freshly prepared in ethanol. For the FRAP assay, solution A (0.0187 g TPTZ in 10 mL ethanol) and solution B (0.338 g sodium acetate in 88.3 mL water with the addition of 1.748 mL acetic acid) were prepared separately and mixed immediately before use. Solutions of FeCl_2_·4H_2_O and FeCl_3_·6H_2_O (1.2 mM) were also prepared freshly before each experiment. The layout of the 96-well plate is shown in Figure 1.

### 2.3. Definition of the Conversion Intervals

Column 1 served as the 100% conversion standard: reduced DPPH for the DPPH assay and FeCl_2_·4H_2_O for the FRAP assay. Column 12 served as the 0% conversion standard: DPPH^●^ radical for the DPPH assay and FeCl_3_·6H_2_O for the FRAP assay.

### 2.4. Application of Tested Samples

Samples (compounds or extracts) were applied to the tested sample field using dilution mode 2, corresponding to a two-thirds serial dilution. Briefly, 150 µL of sample was added to column 2 and 50 µL of ethanol to columns 3–10. Then, 100 µL was sequentially transferred from column 2 to column 11 using a multichannel micropipette, and 100 µL was removed from the last column.

### 2.5. Microplate Setup and Reagent Distribution

50 µL of 0.4 mM DPPH (ethanol) was added to wells A1–C1; 50 µL of 1.2 mM FeCl_2_·4H_2_O (water) to wells D1–F1; 50 µL of 0.4 mM DPPH^●^ (ethanol) to wells A12–C12; and 50 µL of 1.2 mM FeCl_3_·6H_2_O (water) to wells D12–F12.

50 µL of 1.2 mM FeCl_3_·6H_2_O was added to rows D–F (columns 2–11). Background wells (rows G–H, columns 2–11) received 150 µL of 50% (*v*/*v*) ethanol.

All reagents were prepared freshly prior to use.

DPPH reagent: 0.4 mM in ethanol.

FRAP reagent: 10 mL of 6 mM TPTZ (ethanol) mixed with 90 mL of sodium acetate–acetic buffer (0.338 g sodium acetate in 88.5 mL water + 1.478 mL acetic acid).

### 2.6. Assay Initialization

To start the reactions, 150 µL of 0.4 mM DPPH reagent was added to rows A–C, yielding a final DPPH concentration of 0.3 mM. For FRAP, 100 µL of reagent was added to rows D–F, giving 0.3 mM final concentrations of TPTZ and Fe^3+^. Subsequently, 100 µL was removed from all wells to maintain absorbance within the linear range of the Lambert–Beer law.

### 2.7. Incubation and Measurement

The microplate was incubated for 10 min at room temperature, and absorbance was recorded using a Synergy H1 microplate reader (BioTek, Winooski, VT, USA) at 520 nm (DPPH) and 630 nm (FRAP). Optical density (OD) values were corrected for background and expressed as percent conversion relative to 0% and 100% standards.

The DPPH_50_ and FRAP_50_ values represent the compound concentration causing 50% conversion of DPPH^●^ or Fe^3+^ → Fe^2+^, respectively (µM). They were derived from plots of conversion (%) = f(concentration). The Pro-oxidant/Antioxidant Balance Index (PABI) was calculated as follows:Pro-oxidant Antioxidant Balance Index/PABI/ = FRAP_50_/DPPH_50_(1)

### 2.8. Statistical Evaluation

Each experiment was performed in triplicate with eight technical replicates. Results are presented as mean ± standard deviation (SD). Correlation coefficients were calculated using Spearman’s method. Differences were considered statistically significant at * *p* < 0.05. For statistical evaluation, we used MS Excel, Office 16.

## 3. Results

A total of 33 compounds, primarily derived from biotechnological production using plant and explant cultures, were evaluated for their antioxidant and pro-oxidant activities within a concentration range of 0–4.096 mM. Antioxidant activity was quantified by DPPH, and pro-oxidant potential by FRAP. Table 1 reports experimentally determined DPPH_50_ and FRAP_50_ values within the tested concentration range (0–4.096 µM), the Spearman r^2^ coefficients (linearity of the working interval), the PABI (FRAP_50_/DPPH_50_), and CAS identifiers.

The table is ordered by descending antioxidant potency. The strongest activities by DPPH were observed for tannic acid, followed by rhamnetin, α-tocopherol, gallic acid, and others. Across the set, DPPH_50_ and PABI were determined for 16 compounds and FRAP_50_ for 19 compounds. Potency spanned roughly two orders of magnitude (≈70 µM to ≈14 mM by extrapolation). By contrast, PABI varied by about one order of magnitude, from 0.17 (rosmarinic acid; pro-oxidant leaning) to 2.14 (arbutin; antioxidant leaning). Notably, although arbutin shows a relatively high PABI, it falls in the medium-intensity range for both readouts. As we previously reported [26], Trolox (CAS No. 53188-07-1) exhibited a DPPH_50_ of 115.01 ± 1.5 µM and a FRAP_50_ of 171.14 ± 7.9 µM, corresponding to a PABI of 1.49. L-ascorbic acid (CAS No. 50-81-7) showed a DPPH_50_ of 164.06 ± 9.9 µM and a FRAP_50_ of 337.2 ± 0.8 µM, yielding a PABI of 2.06. These reference values provide a comparative framework for interpreting the antioxidant and pro-oxidant characteristics of the compounds examined in the present study.

To contextualize applications and potential impacts on biotechnological production, we considered both balance (PABI) and absolute intensity (DPPH_50_/FRAP_50_). For operational clarity, compounds were classified via parameter FRAP_50_ as highly active (≤300 µM), moderately active (300 µM–1 mM), or low-level activity (>1 mM).

To visualize both activity intensity and balance, we plotted DPPH_50_ (µM) against log(FRAP_50_) for the 15 selected compounds (without Hesperetin) with complete measurements, illustrating relative antioxidant- vs. pro-oxidant-leaning behavior (Figure 2).

Figure 2 shows that most compounds cluster within a narrow linear interval (blue points), indicating a tight relationship between DPPH_50_ and FRAP_50_, which is represented by the following correlation equation: log(FRAP_50_) = 1.2964 × DPPH_50_ + 1.2364 (r^2^ = 0.976). Application-relevant outliers include pro-oxidant-leaning compounds (rosmarinic acid, luteolin; red) and antioxidant-leaning compounds (gallic acid, cyanidin, chlorogenic acid, sinapic acid, arbutin; green). Notably, several outliers also fall in the high-intensity range—rosmarinic acid on the pro-oxidant side and gallic acid, chlorogenic acid, and cyanidin on the antioxidant side.

## 4. Discussion

Our results provide an integrated view of antioxidant and pro-oxidant behavior across 33 plant cell-culture-derived compounds by combining intensity metrics (DPPH_50_, FRAP_50_) with a balance descriptor (PABI). Activities spanned nearly two orders of magnitude, and DPPH_50_ and FRAP_50_ were tightly coupled (strong linear trend), permitting classification into high-, medium-, and low-intensity ranges. Within this framework, several outliers displayed marked antioxidant- or pro-oxidant-leaning profiles with potential application relevance.

This section examines implications for targeted biotechnological production and the potential for pro-oxidant-driven process self-limitation. The targeted biotechnological production of compounds with exceptional properties—either with pronounced pro-oxidant effects, which can underlie antimicrobial or anticancer activity, or with strong antioxidant effects, which are associated with cytoprotective and reparative effects. It is important to recognize that both types of compounds have their respective roles. In the initial phase, the elimination of an etiological agent is desirable (pro-oxidant activity), followed by a recovery phase in which free radicals are scavenged, and homeostasis is stabilized (antioxidant activity). The attractiveness of biotechnological production—whether classical, microbial, or plant-based—therefore largely depends on the desirability and functional profile of the target product. This study provides a framework for differentiating and quantifying antioxidant versus pro-oxidant behaviors. Production of compounds with strong antioxidant activity is well documented [27,28], and targeted pro-oxidant activity has likewise been reported for specific compounds and natural sources [29,30,31].

Another aspect concerns self-limitation of the process, potentially driven by the product’s pro-oxidant properties, with negative implications for yield optimization. Currently, there is limited insight into the extent to which pro-oxidant effects, as reflected by the PABI, may limit the biotechnological production of target compounds. In contrast, model stress conditions are frequently applied to induce the biosynthesis of target metabolites. In industrial settings, strategies such as nitrogen depletion, phosphate starvation, or salinity stress can increase the yield of desired product classes. However, stress also impairs microbial growth and overall productivity of biomass [32], likely due to elevated reactive oxygen species (ROS) under stress conditions [33].

Mechanistically, the pro-oxidant activity of compounds and complex matrices (e.g., plant extracts) is often driven by Fenton-type reactions. These reactions require hydrogen peroxide and the presence of transition metals (e.g., iron, copper) within a reducing environment capable of converting transition metals from a higher to a lower oxidation state. These conditions are present across human, plant, and microbial cells and within culture media. Because culture media necessarily contain transition metals and reductants, hydrogen peroxide is a plausible trigger of Fenton chemistry. To fully understand this phenomenon, a comprehensive mapping of the hydrogen peroxide formation and accumulation, particularly in later cultivation stages, would therefore be essential for process design and control. In this context, several studies provide relevant insight [34,35,36].

Limitations include incomplete FRAP_50_/DPPH_50_ determination for all compounds and the use of in vitro chemical assays that do not capture matrix effects, intracellular redox buffering, or metal speciation in culture media. These constraints motivate integration with cell-based and process-level readouts (e.g., ROS metrics, H_2_O_2_ mapping, growth/productivity indices).

## 5. Conclusions

This pilot study provides an integrated framework for evaluating how the intensity and balance of antioxidant and pro-oxidant activities may impact biotechnological production. We analyzed 33 compounds currently produced via plant cell cultures and quantified activity using DPPH_50_, FRAP_50_, and the PABI. Our findings indicate that the balance between antioxidant and pro-oxidant tendencies (DPPH_50_/FRAP_50_) can guide (i) the selection of compounds for specific applications (cytotoxic vs. cytoprotective use) and (ii) the optimization of cultivation conditions to maximize productivity while minimizing self-limiting oxidative stress.

Future work aims to predict antioxidant/pro-oxidant behavior directly from molecular structure with high accuracy and to expand the dataset to improve model generalizability. Collectively, these insights are expected to guide the design and optimization of biotechnological processes for high-value plant-derived products.

## Figures and Tables

**Figure 1 biotech-14-00091-f001:**
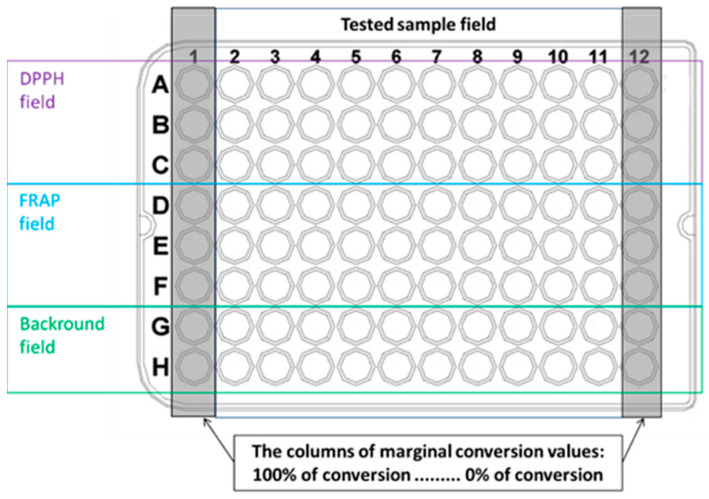
Microplate layout used for DPPH and FRAP assays. The plate includes a tested sample field (wells A2–H11) and columns reserved for standards representing 100% (A1–F1) and 0% (A12–F12) conversion.

**Figure 2 biotech-14-00091-f002:**
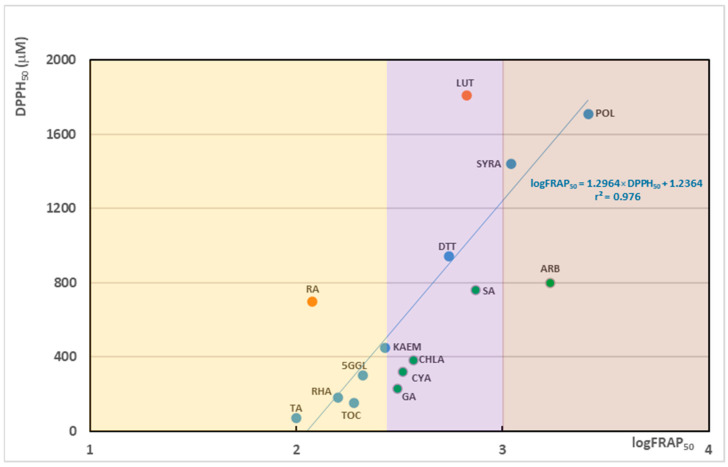
Scatter plot of 15 plant-derived compounds showing DPPH_50_ (µM) versus log(FRAP_50_). Points in blue fall within the correlation interval; orange denote pro-oxidant-leaning compounds; and green represent antioxidant active compounds. Shaded zones indicate intensity ranges as follows: high—log FRAP_50_ ≤ 2.477, transparent orange, medium—log FRAP_50_ is from interval (2.477, 3.0), transparent violet and low—log FRAP_50_ ≥ 3.0, transparent brown. Each point is labeled with the abbreviated compound name: TA/tannic acid/, RA/rosmarinic acid/, RHA/rhamnetin/, α-TOC/α-tocopherol/, 5GGL/1,2,3,4,6-penta-O-galloyl-β-D-glucose/, KAEM/kaempferol/, GA/gallic acid/, CYA/cyanidin/, CHLA/chlorogenic acid/, DTT/1,4-dithiothreitol/, LUT/luteolin/, SA/sinapic acid/, SYRA/syringic acid/, ARB/arbutin/and POL/polydatin/.

**Table 1 biotech-14-00091-t001:** CAS numbers, DPPH_50_, FRAP_50_, Spearman r^2^, and PABI values for 19 compounds from plant and explant cell cultures.

Compound	CAS Number	DPPH_50_ (μM)	r^2^	FRAP_50_ (μM)	r^2^	PABI
Tannic acid	1401-55-4	70 ± 0	0.982	100 ± 11	0.955	1.43
Rhamnetin	90-19-7	180 ± 22	0.991	160 ± 20	0.978	0.89
a-Tocopherol	59-02-9	150 ± 21	0.987	190 ± 18	0.959	1.27
Gallic acid	149-91-7	230 ± 20	0.899	310 ± 31	0.961	1.35
1,2,3,4,6-Penta-O-Galloyl-Β-D-Glucose	14937-32-7	300 ± 42	0.969	210 ± 19	0.957	0.70
Chlorogenic acid	327-97-9	380 ± 44	0.951	370 ± 36	0.972	0.97
Cyanidin. HCl	528-58-5	320 ± 28	0.959	330 ± 21	0.981	1.03
Kaempferol	520-18-3	450 ± 51	0.912	270 ± 30	0.988	0.60
Rosmarinic acid	20283-92-5	700 ± 32	0.984	120 ± 10	0.970	0.17
Sinapic acid	530-59-6	760 ± 81	0.995	740 ± 68	0.986	0.97
Arbutin	497-76-7	800 ± 91	0.976	1710 ± 199	0.964	2.14
1,4-Dithiothreitol	3483-12-3	940 ± 112	0.923	550 ± 69	0.942	0.59
Syringic acid	530-57-4	1440 ± 185	0.985	1100 ± 150	0.986	0.76
Polydatin	65914-17-2	1710 ± 219	0.959	2600 ± 302	0.976	1.52
Luteolin	491-70-3	1810 ± 227	0.947	670 ± 92	0.963	0.37
Hesperetin	520-33-2	6810 ± 792 *	0.924	2210 ± 260	0.893	0.32
3-Hydroxyflavone	577-85-5	over		3520 ± 410	0.961	
Biochanin A	491-80-5	over		5030 ± 0.65 *	0.995	
Scopolamine	51-34-3	over		13,920 ± 1180 *	0.982	

* Value determined by extrapolation. The following compounds exhibited DPPH_50_ and FRAP_50_ values exceeding the assay concentration limit: podophyllotoxin, berberine, paclitaxel, rhein, olivetol, plumbagin, phlorizin, p-coumaric acid, aesculin, hesperidin, purpurin, lapachol, shikonin, and icariin.

## Data Availability

The original contributions presented in the study are included in the article; further inquiries can be directed to the corresponding author.

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
