# Peer review of "Concurrent Analysis of Antioxidant and Pro-Oxidant Activities in Compounds from Plant Cell Cultures"

_biotech, 2025, doi:10.3390/biotech14040091_

Round 1

Reviewer 1 Report

Comments and Suggestions for Authors

The manuscript introduces an innovative attempt to concurrently assess antioxidant and pro-oxidant properties of various plant-derived compounds using modified microplate assays (DPPH and FRAP). The proposed pro-oxidant–antioxidant balance index (PABI = FRAP₅₀ / DPPH₅₀) could be an interesting quantitative descriptor. However, the study in its current form suffers from serious methodological, analytical, and presentation weaknesses, as follows:

1- At the end of the Introduction, the authors do not state the objective(s) of their work; this must be added.
2- Page 3, line 105: “final concentrations ranging from 4,096 to 8 μM” — replace the comma with a decimal point if it is a decimal (→ 4.096 μM)
3- The “Statistical evaluation” section lacks much precision, indicate the program or package used, and p < 0.10 is unusually lax; the standard is p < 0.05 (or 0.01).
4- In the title/caption of Table 1 and on page 3, line 119, it is indicated that there are 30 compounds, but Table 1 lists only 21; please explain and reconcile this discrepancy.
5- FRAP classically measures the reducing ability of antioxidants via the Fe³⁺–TPTZ complex with absorbance at 593 nm. The manuscript reads FRAP at 630 nm and interprets it as “pro-oxidant,” which lacks theoretical or experimental support.
6- Missing details include plate layout, reagent concentrations, blank corrections, calibration, temperature, and incubation control.
7- There is no mention of reference standards or positive/negative controls (e.g., Trolox, Fe²⁺ standards).

Author Response

Comments and Suggestions for Authors

The manuscript introduces an innovative attempt to concurrently assess antioxidant and pro-oxidant properties of various plant-derived compounds using modified microplate assays (DPPH and FRAP). The proposed pro-oxidant–antioxidant balance index (PABI = FRAP₅₀ / DPPH₅₀) could be an interesting quantitative descriptor. However, the study in its current form suffers from serious methodological, analytical, and presentation weaknesses, as follows:

  1. At the end of the Introduction, the authors do not state the objective(s) of their work; this must be added.

The primary objective of this pilot study is to evaluate potential differences in the balance between antioxidant and pro-oxidant activities among compounds produced through in vitro plant biotechnological systems. Elucidating these differences may provide valuable insights into their prospective applicability, while also indicating that compounds exhibiting a predominance of pro-oxidant activity could constitute intrinsic self-limiting factors within the biosynthetic processes of their own production systems.

  1. Page 3, line 105: “final concentrations ranging from 4,096 to 8 μM” — replace the comma with a decimal point if it is a decimal (→ 4.096 μM).

It has been corrected.

  1. The “Statistical evaluation” section lacks much precision, indicate the program or package used, and p < 0.10 is unusually lax; the standard is p < 0.05 (or 0.01).

It has been corrected.

  1. In the title/caption of Table 1 and on page 3, line 119, it is indicated that there are 30 compounds, but Table 1 lists only 21; please explain and reconcile this discrepancy.

The number of tested compounds, including those with values exceeding the assay concentration limits, has been corrected in both the text and the Table 1 caption to ensure consistency.

  1. FRAP classically measures the reducing ability of antioxidants via the Fe³⁺–TPTZ complex with absorbance at 593 nm. The manuscript reads FRAP at 630 nm and interprets it as “pro-oxidant,” which lacks theoretical or experimental support.

The absorption spectrum of the FRAP reaction product exhibits a relatively flat profile in the wavelength range of 580–700 nm. The wavelength of 630 nm has been consistently used in our assays to minimize potential color interference from tested compounds and extract matrices. This methodological approach has been applied and validated in several studies from our research group over the past decade, demonstrating its reliability and reproducibility under our experimental conditions.

  1. Missing details include plate layout, reagent concentrations, blank corrections, calibration, temperature, and incubation control.

A detailed description of the methodology has been added, including the plate layout, reagent concentrations, blank corrections, calibration procedures, and conditions for temperature and incubation control.

  1. There is no mention of reference standards or positive/negative controls (e.g., Trolox, Fe²⁺ standards).

We implemented a paragraph to the Results section: As we previously reported [26], Trolox (CAS No. 53188-07-1) exhibited a DPPH₅₀ of 115.01 ± 1.5 µM and a FRAP₅₀ of 171.14 ± 7.9 µM, corresponding to a PABI index of 1.49. L-ascorbic acid (CAS No. 50-81-7) showed a DPPH₅₀ of 164.06 ± 9.9 µM and a FRAP₅₀ of 337.2 ± 0.8 µM, yielding a PABI index of 2.06. These reference values provide a comparative framework for interpreting the antioxidant and pro-oxidant characteristics of the compounds examined in the present study.

Reviewer 2 Report

Comments and Suggestions for Authors

Authors provided intriguing manuscript regarding anti/pro- oxidative properties of various plant secondary metabolites. The topic, chosen methods and results show another eventually possible approach for evaluation of products from in vitro cultures and other sources. Unfortunately, there are some typos and issues that should be edited and/or further explained in the revision.

Abstract

Line 25 - "Seventeen compounds..." - this count does not correspond to the rest of the text.

Introduction

Line 56 - missing space behind dash. Also control and edit use of hyphen/dash in further text.

Lines 62-83 - please divide this long paragraph to more shorter ones for better readability and more sources could be used for the description of secondary metabolites as well. Moreover, I would not also classified betalains to the group of alkaloids despite their origin and nitrogen bounded in cycle. Lobeline and piperidine could be also together; generally, subgroups of secondary metabolites should be stated first and specific compounds as examples after them.

Line 72 - brassinolide.

Materials and Methods

Line 86 - what was the concentration of ethanol? It was used as solvent (and possible blank as I suppose) for all standards? 

Line 97 - specification of used multiplate reader should be stated.

Line 105 and 123 - 4.096.

Line 107 - please control the text for redundant space between words. Here, it looks that there is one between "...by  adding...".

Results

Line 123 - the concentration range does not correspond to the previous section (4.096 to 8 μM, line 105).

Line 127 - "...18 compounds,..." - there are only 16 compounds in Table 1 with DPPH/PABI results and 19 for FRAP.

Lines 129, 130 - according to the text, PABI is calculated as FRAP50/DPPH50 (e.g. Line 111, thus ratio of pro-oxidant/antioxidant activity (e.g. Line 121). I would say that compounds with values under 1.0 should act more like antioxidant then (higher denominator than numerator) and the conclusions regarding specific compounds would be reversed then. Please, provide explanation for better clarification here.

Lines 135 - the term "active" should be also described here, not only in legend for Figure 1.

Table 1 - typo in name of kaempferol; commas instead of dots in PABI of tocopherol and cyanidin HCl. 5GALGLUC has no CAS number, but if this chemical is from Merck/Sigma as authors stated, could be at least its product number provided in their response? 

Lines 144/145 - information regarding concentration limit for these compounds should be in main text. Authors could also discuss alternative methods for activity measurement for these metabolites in next section (4.). Moreover, there are 30 stated standards, but results of purpurin, lapachol, shikonin are not showed.

Figure 1 - there are only 15 points and 14 names of compounds.

Discussion

As mentioned above, possible alternative methods could be mentioned here as well (if applicable).

Line 172 - "...across 30 plant cell-culture-derived compounds..." - data was provided only for 16/19 of them.

Line 211 - two dots.

Author Response

Authors provided intriguing manuscript regarding anti/pro- oxidative properties of various plant secondary metabolites. The topic, chosen methods and results show another eventually possible approach for evaluation of products from in vitro cultures and other sources. Unfortunately, there are some typos and issues that should be edited and/or further explained in the revision.

Abstract

Line 25 - "Seventeen compounds..." - this count does not correspond to the rest of the text.

The inconsistency in compound count has been corrected throughout the manuscript.

Introduction

Line 56 - missing space behind dash. Also control and edit use of hyphen/dash in further text.

It has been corrected.

Lines 62-83 - please divide this long paragraph to more shorter ones for better readability and more sources could be used for the description of secondary metabolites as well. Moreover, I would not also classified betalains to the group of alkaloids despite their origin and nitrogen bounded in cycle. Lobeline and piperidine could be also together; generally, subgroups of secondary metabolites should be stated first and specific compounds as examples after them.

The paragraph has been divided into several shorter sections to improve readability. Betalains were removed from the alkaloid group, and the order of subgroups and representative compounds has been adjusted as suggested.

Line 72 - brassinolide.

Corrected.

Materials and Methods

Line 86 - what was the concentration of ethanol? It was used as solvent (and possible blank as I suppose) for all standards?

The concentration of ethanol was specified as 96% (v/v), and it was used as the solvent and blank throughout the study.

Line 97 - specification of used multiplate reader should be stated.

The full specification of the microplate reader has been added to the Methods section.

Line 105 and 123 - 4.096.

Corrected.

Line 107 - please control the text for redundant space between words. Here, it looks that there is one between "...by  adding...".

It has been checked and corrected.

Results

Line 123 - the concentration range does not correspond to the previous section (4.096 to 8 μM, line 105).

The concentration range has been revised to ensure consistency.

Line 127 - "...18 compounds,..." - there are only 16 compounds in Table 1 with DPPH/PABI results and 19 for FRAP.

The text has been updated to accurately reflect the number of compounds analyzed in each assay.

Lines 129, 130 - according to the text, PABI is calculated as FRAP50/DPPH50 (e.g. Line 111, thus ratio of pro-oxidant/antioxidant activity (e.g. Line 121). I would say that compounds with values under 1.0 should act more like antioxidant then (higher denominator than numerator) and the conclusions regarding specific compounds would be reversed then. Please, provide explanation for better clarification here.

The PABI index is defined as the ratio FRAP₅₀ / DPPH₅₀, representing the relative concentrations required to achieve 50% conversion in the respective assays. It is important to note that this ratio reflects an inverse relationship between concentration and activity, a lower concentration indicates higher efficacy. Accordingly, when a compound exhibits a PABI value below 1, it means that FRAP₅₀ was reached at a lower concentration than DPPH₅₀, indicating a predominance of pro-oxidant activity. In contrast, a PABI value above 1 suggests that antioxidant activity predominates.

Lines 135 - the term "active" should be also described here, not only in legend for Figure 1.

The definition of the term “active” has been added in the main text for clarity and consistency with the figure.

Table 1 - typo in name of kaempferol; commas instead of dots in PABI of tocopherol and cyanidin.

Corrected.

HCl. 5GALGLUC has no CAS number, but if this chemical is from Merck/Sigma as authors stated, could be at least its product number provided in their response?

The product and CAS numbers have been provided.

Lines 144/145 - information regarding concentration limit for these compounds should be in main text. The information regarding the concentration limits of the tested compounds has been added to the main text.

Authors could also discuss alternative methods for activity measurement for these metabolites in next section (4.).

We appreciate the reviewer’s suggestion to discuss alternative methods for activity measurement. However, this lies beyond the primary scope of the present study. To the best of our knowledge, no other methodology currently enables simultaneous assessment of both antioxidant and pro-oxidant activities within a single experimental setup, as established in our previous work (Maliar et al., 2023). Moreover, separate measurements of these two activities are often less reliable and not directly comparable under identical experimental conditions.

Moreover, there are 30 stated standards, but results of purpurin, lapachol, shikonin are not shown.

The text and tables have been revised to ensure consistency between the number of stated standards and the results presented. From 33 tested compounds, only 16 expressed measurable value of DPPH50 and 19 value of FRAP50, see for detail Table 1. Compound Hesperetin has a value of DPPH50 calculated by extrapolation and it is not be included in chart on Figure 2.

Figure 1 - there are only 15 points and 14 names of compounds.

The figure has been corrected, and the abbreviations for all compounds have been added to the figure legend for clarity.

Discussion

As mentioned above, possible alternative methods could be mentioned here as well (if applicable).

Addressed in the text as explained in the previous response.

Line 172 - "...across 30 plant cell-culture-derived compounds..." - data was provided only for 16/19 of them.

The statement has been revised to accurately correspond to the number of compounds with available data.

Line 211 - two dots.

Corrected.

Round 2

Reviewer 1 Report

Comments and Suggestions for Authors

The authors have made all the necessary corrections and provided satisfactory answers to all the questions. I recommend the publication of this paper in its current form.

Author Response

Reviewer 1:

Without comments and questions.

Reviewer 2 Report

Comments and Suggestions for Authors

Authors thoroughly edited their manuscript and sufficiently answered the questions regarding some unclear issues in the first version. Thus, there are only few minor flaws that should be corrected before accepting.

Lines 62-84 - there is no dividing to shorter paragraphs in sent revised version. Similarly, sequence of secondary metabolites/specific compounds remained same in some cases, e. g. quinones. Please, sort mentioned compounds behind the appropriate subgroup of metabolites (anthraquinones - aloe-emodin, emodin, chrysophanol, rhein; naphtoquinones - plumbagin, shikonin, juglone; benzoquinones - thymoquinone; etc.)

Lines 95 - (v/v).

Line 96 - 5-galloylglucose is further in text described as 1,2,3,4,6-penta-O-galloyl-ß-D-glucose. If it had not been requested by other reviewers, I would have stuck with the shorter name or the abbreviation shown under Figure 2 (5GGL). 

Lines 200-205 - control dashes/hyphens and missing spaces in legend under Figure 2.

Lines 217 - regarding the Table 1/Figure 2, there are only 16 compounds with provided results of antioxidant and pro-oxidant behavior. The rest exceeding concentration limit, so it does not correspond to the number of compounds with available data. 

Author Response

Lines 62-84 - there is no dividing to shorter paragraphs in sent revised version. Similarly, sequence of secondary metabolites/specific compounds remained same in some cases, e. g. quinones. Please, sort mentioned compounds behind the appropriate subgroup of metabolites (anthraquinones - aloe-emodin, emodin, chrysophanol, rhein; naphtoquinones - plumbagin, shikonin, juglone; benzoquinones - thymoquinone; etc.)

It has been corrected in according with suggestions.

Lines 95 - (v/v).

It has been corrected.

Line 96 - 5-galloylglucose is further in text described as 1,2,3,4,6-penta-O-galloyl-ß-D-glucose. If it had not been requested by other reviewers, I would have stuck with the shorter name or the abbreviation shown under Figure 2 (5GGL).

It has been corrected in according with our decision, the first time in the manuscript text as well in teble should be full name.

Lines 200-205 - control dashes/hyphens and missing spaces in legend under Figure 2.

Completely re-processed and checked.

Lines 217 - regarding the Table 1/Figure 2, there are only 16 compounds with provided results of antioxidant and pro-oxidant behavior. The rest exceeding concentration limit, so it does not correspond to the number of compounds with available data.

A total of 33 compounds were tested, of which 19 are listed in the table with quantified DPPH₅₀ or FRAP₅₀ values. The remaining 14 compounds are listed below the table with an explanation that their tested activities were outside the concentration range—this is also considered an experimental finding. Only 15 compounds with DPPH₅₀ and FRAP₅₀ values determined by interpolation were included in the graph in Figure 2. The compound Hesperetin was intentionally excluded from the graph, as its DPPH₅₀ value was determined by extrapolation. We believe that such a decision lies within the authors’ discretion.